# Spatiotemporal Residual Networks
# for Video Action Recognition

**Christoph Feichtenhofer**
Graz University of Technology
feichtenhofer@tugraz.at

**Axel Pinz**
Graz University of Technology
axel.pinz@tugraz.at

**Richard P. Wildes**
York University, Toronto
wildes@cse.yorku.ca

## Abstract

Two-stream Convolutional Networks (ConvNets) have shown strong performance for human action recognition in videos. Recently, Residual Networks (ResNets) have arisen as a new technique to train extremely deep architectures. In this paper, we introduce spatiotemporal ResNets as a combination of these two approaches. Our novel architecture generalizes ResNets for the spatiotemporal domain by introducing residual connections in two ways. First, we inject residual connections between the appearance and motion pathways of a two-stream architecture to allow spatiotemporal interaction between the two streams. Second, we transform pretrained image ConvNets into spatiotemporal networks by equipping them with learnable convolutional filters that are initialized as temporal residual connections and operate on adjacent feature maps in time. This approach slowly increases the spatiotemporal receptive field as the depth of the model increases and naturally integrates image ConvNet design principles. The whole model is trained end-to-end to allow hierarchical learning of complex spatiotemporal features. We evaluate our novel spatiotemporal ResNet using two widely used action recognition benchmarks where it exceeds the previous state-of-the-art.

## 1 Introduction

Action recognition in video is an intensively researched area, with many recent approaches focused on application of Convolutional Networks (ConvNets) to this task, e.g. [13, 20, 26]. As actions can be understood as spatiotemporal objects, researchers have investigated carrying spatial recognition principles over to the temporal domain by learning local spatiotemporal filters [13, 25, 26]. However, since the temporal domain arguably is fundamentally different from the spatial one, different treatment of these dimensions has been considered, e.g. by incorporating optical flow networks [20], or modelling temporal sequences in recurrent architectures [4, 18, 19].

Since the introduction of the "AlexNet" architecture [14] in the 2012 ImageNet competition, ConvNets have dominated state-of-the-art performance across a variety of computer vision tasks, including object-detection, image segmentation, image classification, face recognition, human pose estimation and tracking. In conjunction with these advances as well as the evolution of network architectures, several design best practices have emerged [8, 21, 23, 24]. First, information bottlenecks should be avoided and the representation size should gently decrease from the input to the output as the number of feature channels increases with the depth of the network. Second, the receptive field at the end of the network should be large enough that the processing units can base operations on larger regions of the input. This functionality can be achieved by stacking many small filters or using large filters in the network; notably, the first choice can be implemented with fewer operations (faster, fewer parameters) and also allows inclusion of more nonlinearities. Third, dimensionality reduction ($1 \times 1$ convolutions) before spatially aggregating filters (e.g. $3 \times 3$) is supported by the fact that outputs of neighbouring filters are highly correlated and therefore these activations can be reduced before aggregation [23]. Fourth, spatial factorization into asymmetric filters can even further reduce computational cost and

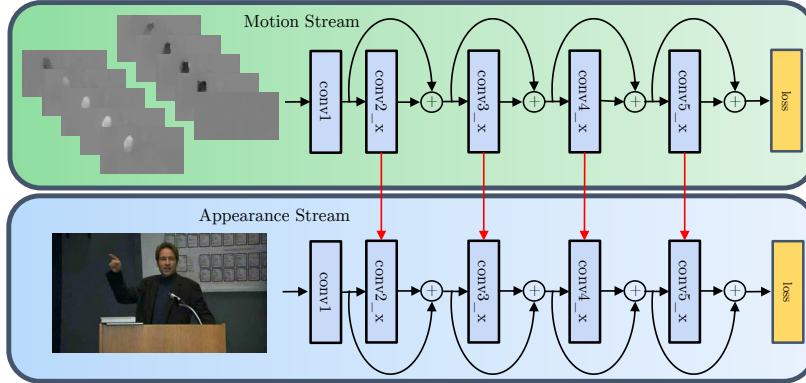

Figure 1: Our method introduces residual connections in a two-stream ConvNet model [20]. The two networks separately capture spatial (appearance) and temporal (motion) information to recognize the input sequences. We do not use residuals from the spatial into the temporal stream as this would bias both losses towards appearance information.

ease the learning problem. Fifth, it is important to normalize the responses of each feature channel within a batch to reduce internal covariate shift [11]. The last architectural guideline is to use residual connections to facilitate training of very deep models that are essential for good performance [8]. We carry over these good practices for designing ConvNets in the image domain to the video domain by converting the $1 \times 1$ convolutional dimensionality mapping filters in ResNets to temporal filters. By stacking several of these transformed temporal filters throughout the network we provide a large receptive field for the discriminative units at the end of the network. Further, this design allows us to convert spatial ConvNets into spatiotemporal models and thereby exploits the large amount of training data from image datasets such as ImageNet.

We build on the two-stream approach [20] that employs two separate ConvNet streams, a spatial *appearance* stream, which achieves state-of-the-art action recognition from RGB images and a temporal *motion* stream, which operates on optical flow information. The two-stream architecture is inspired by the two-stream hypothesis from neuroscience [6] that postulates two pathways in the visual cortex: The ventral pathway, which responds to spatial features such as shape or colour of objects, and the dorsal pathway, which is sensitive to object transformations and their spatial relationship, as e.g. caused by motion. We extend two-stream ConvNets in the following ways. First, motivated by the recent success of residual networks (ResNets) [8] for numerous challenging recognition tasks on datasets such as ImageNet and MS COCO, we apply ResNets to the task of human action recognition in videos. Here, we initialize our model with pre-trained ResNets for image categorization [8] to leverage a large amount of image-based training data for the action recognition task in video. Second, we demonstrate that injecting residual connections between the two streams (see Fig. 1) and jointly fine-tuning the resulting model achieves improved performance over the two-stream architecture. Third, we overcome limited temporal receptive field size in the original two-stream approach by extending the model over time. We convert convolutional dimensionality mapping filters to temporal filters that provide the network with *learnable* residual connections over time. By stacking several of these temporal filters and sampling the input sequence at large temporal strides (i.e. skipping frames), we enable the network to operate over large temporal extents of the input. To demonstrate the benefits of our proposed spatiotemporal ResNet architecture, it has been evaluated on two standard action recognition benchmarks where it greatly boosts the state-of-the-art.

## 2 Related work

Approaches for action recognition in video can largely be divided into two categories: Those that use hand-crafted features with decoupled classifiers and those that jointly learn features and classifier. Our work is related to the latter, which is outlined in the following.

Several approaches have been presented for spatiotemporal feature learning. Unsupervised learning techniques have been applied by stacking ISA or convolutional gated RBMs to learn spatiotemporal features for action recognition [16, 25]. In other work, spatiotemporal features are learned by extending 2D ConvNets into time by stacking consecutive video frames [12]. Yet another study compared several approaches to extending ConvNets into the temporal domain, but with rather disappointing results [13]: The architectures were not particularly sensitive to temporal modelling,

with a slow fusion model performing slightly better than early and late fusion alternatives; moreover, similar levels of performance were achieved by a purely spatial network. The recently proposed C3D approach learns 3D ConvNets on a limited temporal support of 16 frames and all filter kernels having size 3×3×3 [26]. The network structure is similar to earlier deep spatial networks [21].

Another research branch has investigated combining image information in network architectures across longer time periods. A comparison of temporal pooling architectures suggested that temporal pooling of convolutional layers performs better than slow, local, or late pooling, as well as temporal convolution [18]. That work also considered ordered sequence modelling, which feeds ConvNet features into a recurrent network with Long Short-Term Memory (LSTM) cells. Using LSTMs, however, did not yield an improvement over temporal pooling of convolutional features. Other work trained an LSTM on human skeleton sequences to regularize another LSTM that uses an Inception network for frame-level descriptor input [17]. Yet other work uses a multilayer LSTM to let the model attend to relevant spatial parts in the input frames [19]. Further, the inner product of a recurrent model has been replaced with a 2D convolution and thereby converts the fully connected hidden layers in a GRU-RNN to 2D convolutional operations [1]. That approach takes advantage of the local spatial similarity in images; however, it only yields a minor increase over their baseline, which is a two-stream VGG-16 ConvNet [21] used as the input to their convolutional RNN. Finally, three recent approaches for action recognition apply ConvNets as follows: In [2] dynamic images are created by weighted averaging of video frames over time; [31] captures the transformation of ConvNet features from the beginning to the end of the video with a Siamese architecture; and [5] introduces a spatiotemporal convolutional fusion layer between the streams of a two-stream architecture.

Notably, the most closely related work to ours (and to several of those above) is the two-stream ConvNet architecture [20]. That approach first decomposes video into spatial and temporal components by using RGB and optical flow frames. These components are fed into separate deep ConvNet architectures to learn spatial as well as temporal information about the appearance and movement of the objects in a scene. Each stream initially performs video recognition on its own and for final classification, softmax scores are combined by late fusion. To date, this approach is the most effective approach of applying deep learning to action recognition, especially with limited training data. In our work we directly convert image ConvNets into 3D architectures and show greatly improved performance over the two-stream baseline.

## 3 Technical approach

### 3.1 Two-Stream residual networks

As our base representation we use deep ResNets [8, 9]. These networks are designed similarly to the VGG networks [21], with small 3×3 spatial filters (except at the first layer), and similar to the Inception networks [23], with 1×1 filters for learned dimensionality reduction and expansion. The network sees an input of size 224×224 that is reduced five times in the network by stride 2 convolutions followed by a global average pooling layer of the final 7×7 feature map and a fully-connected classification layer with softmax. Each time the spatial size of the feature map changes, the number of features is doubled to avoid tight bottlenecks. Batch normalization [11] and ReLU [14] are applied after each convolution; the network does not use hidden fc, dropout, or max-pooling (except immediately after the first layer). The residual units are defined as [8, 9]:

$$\mathbf{x}_{l+1} = f\left(\mathbf{x}_l + \mathcal{F}(\mathbf{x}_l; \mathcal{W}_l)\right), \tag{1}$$

where $\mathbf{x}_l$ and $\mathbf{x}_{l+1}$ are input and output of the $l$-th layer, $\mathcal{F}$ is a nonlinear residual mapping represented by convolutional filter weights $\mathcal{W}_l = \{\mathrm{W}_{l,k}|_{1 \leq k \leq K}\}$ with $K \in \{2, 3\}$ and $f \equiv \mathrm{ReLU}$ [9]. A key advantage of residual units is that their skip connections allow direct signal propagation from the first to the last layer of the network. Especially during backpropagation this arrangement is advantageous: Gradients are propagated directly from the loss layer to any previous layer while skipping intermediate weight layers that have potential to trigger vanishing or deterioration of the gradient signal.

We also leverage the two-stream architecture [20]. For both streams, we use the ResNet-50 model [8] pretrained on the ImageNet dataset and replace the last (classification) layer according to the number of classes in the target dataset. The filters in the first layer of the motion stream are further modified by replicating the three RGB filter channels to a size of $2L = 20$ for operating over the horizontal and vertical optical flow stacks, each of which has a stack of $L = 10$ frames. This tack allows us to exploit the availability of a large amount of annotated training data for both streams.

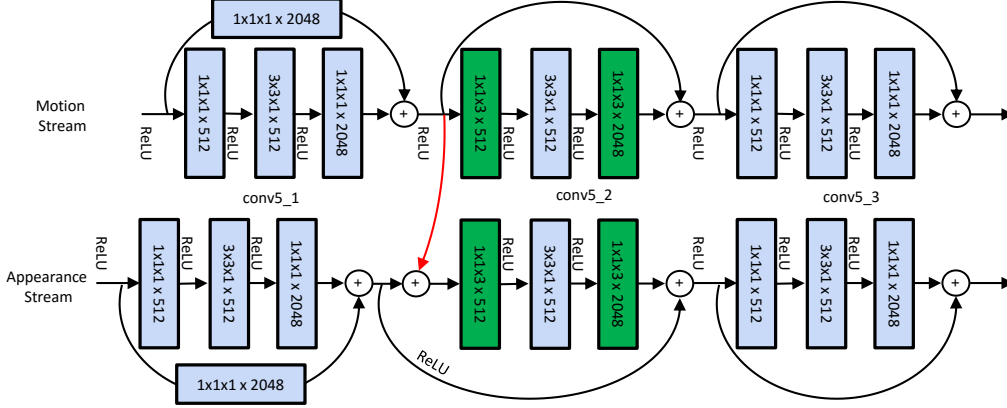

Figure 2: The conv5_x residual units of our architecture. A residual connection (highlighted in red) between the two streams enables motion interactions. The second residual unit, conv5_2 also includes temporal convolutions (highlighted in green) for learning abstract spacetime features.

A drawback of the two-stream architecture is that it is unable to spatiotemporally register appearance and motion information. Thus, it is not able to represent what (captured by the spatial stream) moves in which way (captured by the temporal stream). Here, we remedy this deficiency by letting the network learn such *spatiotemporal* cues at several spatiotemporal scales. We enable this interaction by introducing residual connections between the two streams. Just as there can be various types of shortcut connections in a ResNet, there are several ways the two streams can be connected. In preliminary experiments we found that direct connections between identical layers of the two streams led to an increase in validation error. Similarly, bidirectional connections increased the validation error significantly. We conjecture that these results are due to the large change that the signal of one network stream undergoes after injecting a fusion signal from the other stream. Therefore, we developed a more subtle alternative solution based on additive interactions, as follows.

**Motion Residuals.**   We inject a skip connection from the motion stream to the appearance stream's residual unit. To enable learning of spatiotemporal features at all possible scales, this modification is applied before the second residual unit at each spatial resolution of the network (indicated by "skip-stream" in Table 1), as exemplified by the connection at the conv5_x layers in Fig. 2. Formally, the corresponding appearance stream's residual units (1) are modified according to

$$\hat{\mathbf{x}}_{l+1}^{a} = f(\mathbf{x}_{l}^{a}) + \mathcal{F}\Big(\mathbf{x}_{l}^{a} + f(\mathbf{x}_{l}^{m}), \mathcal{W}_{l}^{a}\Big), \tag{2}$$

where $\mathbf{x}_{l}^{a}$ is the input of the $l$-th layer appearance stream, $\mathbf{x}_{l}^{m}$ the input of the $l$-th layer motion stream and $\mathcal{W}_{l}^{a}$ are the weights of the $l$-th layer residual unit in the appearance stream. For the gradient on the loss function $\mathcal{L}$ in the backward pass the chain rule yields

$$\frac{\partial \mathcal{L}}{\partial \mathbf{x}_{l}^{a}} = \frac{\partial \mathcal{L}}{\partial \hat{\mathbf{x}}_{l+1}^{a}} \frac{\partial \hat{\mathbf{x}}_{l+1}^{a}}{\partial \mathbf{x}_{l}^{a}} = \frac{\partial \mathcal{L}}{\partial \hat{\mathbf{x}}_{l+1}^{a}} \left( \frac{\partial f(\mathbf{x}_{l}^{a})}{\partial \mathbf{x}_{l}^{a}} + \frac{\partial}{\partial \mathbf{x}_{l}^{a}} \mathcal{F}\Big(\mathbf{x}_{l}^{a} + f(\mathbf{x}_{l}^{m}), \mathcal{W}_{l}^{a}\Big) \right) \tag{3}$$

for the appearance stream and similarly for the motion stream

$$\frac{\partial \mathcal{L}}{\partial \mathbf{x}_{l}^{m}} = \frac{\partial \mathcal{L}}{\partial \mathbf{x}_{l+1}^{m}} \frac{\partial \mathbf{x}_{l+1}^{m}}{\partial \mathbf{x}_{l}^{m}} + \frac{\partial \mathcal{L}}{\partial \hat{\mathbf{x}}_{l+1}^{a}} \frac{\partial}{\partial \mathbf{x}_{l}^{a}} \mathcal{F}\Big(\mathbf{x}_{l}^{a} + f(\mathbf{x}_{l}^{m}), \mathcal{W}_{l}^{a}\Big), \tag{4}$$

where the first additive term of (4) is the gradient at the $l$-th layer in the motion stream and the second term accumulates gradients from the appearance stream. Thus, the residual connection between the streams backpropagates gradients from the appearance stream into the motion stream.

## 3.2   Convolutional residual connections across time

Spatiotemporal coherence is an important cue when working with time varying visual data and can be exploited to learn general representations from video in an unsupervised manner [7]. In that case, temporal smoothness is an important property and is enforced by requiring features to vary slowly with respect to time. Further, one can expect that in many cases a ConvNet is capturing similar features across time. For example, an action with repetitive motion patterns such as "Hammering" would trigger similar features for the appearance and motion stream over time. For such cases the use of temporal residual connections would make perfect sense. However, for cases where the

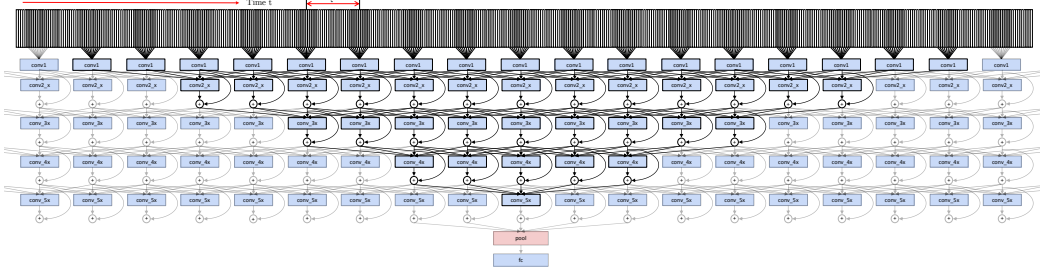

Figure 3: The temporal receptive field of a single neuron at the fifth meta layer of our motion network stream is highlighted. $\tau$ indicates the temporal stride between inputs. The outputs of conv5_3 are max-pooled in time and fed to the fully connected layer of our ST-ResNet*.

appearance or the instantaneous motion pattern varies over time, a residual connection would be suboptimal for discriminative learning, since the sum operation corresponds to a low-pass filtering over time and would smooth out potentially important high-frequency temporal variation of the features. Moreover, backpropagation is unable to compensate for that deficit since at a sum layer all gradients are distributed equally from output to input connections.

Based on the above observations, we developed a novel approach to temporal residual connections that builds on the ConvNet design guidelines of chaining small [21] asymmetric [10, 23] filters, noted in Sec. 1. We extend the ResNet architecture with temporal convolutions by *transforming* spatial dimensionality mapping filters in the residual paths to temporal filters. This allows the straightforward use of standard two-stream ConvNets that have been pre-trained on large-scale datasets e.g. to leverage the massive amounts of training data from the ImageNet challenge. We initialize the temporal weights as residual connections across time and let the network *learn* to best discriminate image dynamics via backpropagation. We achieve this by replicating the learned spatial $1\times1$ dimensionality mapping kernels in pretrained ResNets across time. Given the pretrained spatial weights, $\mathbf{w}_l \in \mathbb{R}^{1\times1\times C}$, temporal filters, $\hat{\mathbf{w}}_l \in \mathbb{R}^{1\times1\times T'\times C}$, are initialized according to

$$\hat{\mathbf{w}}_l(i,j,t,c) = \frac{\mathbf{w}_l(i,j,c)}{T'}, \forall t \in [1, T'], \tag{5}$$

and subsequently refined via backpropagation. In (5), division by $T'$ serves to average feature responses across time. We transform filters from both the motion and the appearance ResNets accordingly. Hence, the temporal filters are able to learn the temporal evolution of the appearance and motion features and, moreover, by stacking such filters as the depth of the network increases complex spatiotemporal relationships can be modelled.

### 3.3 Proposed architecture

Our overall architecture (used for each stream) is summarized in Table 1. The underlying network used is a 50 layer ResNet [8]. Each filtering operation is followed by batch normalization [11] and halfway rectification (ReLU). In the columns we show "metalayers" which share the same output size. From left to right, top to bottom, the first row shows the convolutional and pooling building blocks, with the filter and pooling size shown as $(W \times H \times T, C)$, denoting width, height, temporal extent and number of feature channels, resp. Brackets outline residual units equipped with skip connections. In the last two rows we show the output size of these metalayers as well as the receptive field on which they operate. One observes that the temporal receptive field is modulated by the temporal stride $\tau$ between the input chunks. For example, if the stride is set to $\tau = 15$ frames, a unit at conv5_3 sees a window of $17 * 15 = 255$ frames on the input video; see. Fig. 3. The pool5 layer receives multiple spatiotemporal features, where the spatial $7 \times 7$ features are averaged as in [8] and the temporal features are max-pooled within a window of 5, with each of these seeing a window of 705 frames at the input. The pool5 output is classified by a fully connected layer of size $1 \times 1 \times 1 \times 2048$; note that this passes several temporally max-pooled chunks to the softmax log-loss layer afterwards. For videos with less than 705 frames we reduce the stride between temporal inputs and for extremely short videos we symmetrically pad the input over time.

**Sub-batch normalization.** Batch normalization [11] subtracts from all activations the batchwise mean and divides by their variance. These moments are estimated by averaging over spatial locations and multiple images in the batch. After batch normalization a learned, channel-specific affine transformation (scaling and bias) is applied. The noisy bias/variance estimation replaces the need

| Layers | conv1 | pool1 | conv2_x | conv3_x | conv4_x | conv5_x | pool5 |
|---|---|---|---|---|---|---|---|
| Blocks | 7×7×1, 64 | 3×3×1 max stride 2 | [1×1×1, 64; 3×3×1, 64; 1×1×1, 256]<br>skip-stream<br>[1×1×3, 64; 3×3×1, 64; 1×1×3, 256]<br>[1×1×1, 64; 3×3×1, 64; 1×1×1, 256] | [1×1×1, 128; 3×3×1, 128; 1×1×1, 512]<br>skip-stream<br>[1×1×3, 128; 3×3×1, 128; 1×1×3, 512]<br>[1×1×1, 128; 3×3×1, 128; 1×1×1, 512] ×2 | [1×1×1, 256; 3×3×1, 256; 1×1×1, 1024]<br>skip-stream<br>[1×1×3, 256; 3×3×1, 256; 1×1×3, 1024]<br>[1×1×1, 256; 3×3×1, 256; 1×1×1, 1024] ×4 | [1×1×1, 512; 3×3×1, 512; 1×1×1, 2048]<br>skip-stream<br>[1×1×3, 512; 3×3×1, 512; 1×1×3, 2048]<br>[1×1×1, 512; 3×3×1, 512; 1×1×1, 2048] | 7×7×1 avg, 1×1×5 max stride 2 |
| Output size | 112×112×11 | 56×56×11 | 56×56×11 | 28×28×11 | 14×14×11 | 7×7×11 | 1×1×4 |
| Recept. Field | 7×7×1 | 11×11×1 | 35×35×5$\tau$ | 99×99×9$\tau$ | 291×291×13$\tau$ | 483×483×17$\tau$ | 675 × 675 × 47$\tau$ |

Table 1: Spatiotemporal ResNet architecture used in both ConvNet streams. The metalayers are shown in the columns with their building blocks showing the convolutional filter dimensions ($W \times H \times T, C$) in brackets. Each building block shown in brackets also has a skip connection to the block below and skip-stream denotes a residual connection from the motion to the appearance stream, e.g., see Fig. 2 for the conv5_2 building block. Stride 2 downsampling is performed by conv1, pool1, conv3_1, conv4_1 and conv5_1. The output and receptive field size of these layers is shown below. For both streams, the pool5 layer is followed by a $1 \times 1 \times 1 \times 2048$ fully connected layer, a softmax and a loss.

for dropout regularization [8, 24]. We found that lowering the number of samples used for batch normalization can further improve the generalization performance of the model. For example, for the appearance stream we use a low batch size of 4 for moment estimation during training. This practice strongly supports generalization of the model and nontrivially increases validation accuracy ($\approx$4% on UCF101). Interestingly, in comparison to this approach, using dropout after the classification layer (e.g. as in [24]) decreased validation accuracy of the appearance stream. Note that only the batchsize for normalizing the activations is reduced; the batch size in stochastic gradient descent is unchanged.

## 3.4 Model training and evaluation

Our method has been implemented in MatConvNet [28] and we share our code and models at https://github.com/feichtenhofer/st-resnet. We train our model in three optimization steps with the parameters listed in Table 2.

| Training phase | SGD batch size | Bnorm batch size | Learning Rate (#Iterations) | Temporal chunks / stride $\tau$ |
|---|---|---|---|---|
| Motion stream | 256 | 86 | $10^{-2}(30K)$, $10^{-3}(10K)$ | $1 / \tau = 1$ |
| Appearance stream | 256 | 8 | $10^{-2}(10K)$, $10^{-3}(10K)$ | $1 / \tau = 1$ |
| ST-ResNet | 128 | 4 | $10^{-3}(30K)$, $10^{-4}(30K)$, $10^{-5}(20K)$ | $5 / \tau \in [5, 15]$ |
| ST-ResNet* | 128 | 4 | $10^{-4}(2K)$, $10^{-5}(2K)$ | $11 / \tau \in [1, 15]$ |

Table 2: Parameters for the three training phases of our model

**Motion and appearance streams.** First, each stream is trained similar to [20] using Stochastic Gradient Descent (SGD) with momentum of 0.9. We rescale all videos by keeping the aspect ratio and resizing the smallest side of a frame to 256. The motion network uses optical flow stacking with $L = 10$ frames and is trained for $30K$ iterations with a learning rate of $10^{-2}$ followed by $10K$ iterations at a learning rate of $10^{-3}$. At each iteration, a batch of 256 samples is constructed by randomly sampling a single optical flow stack from a video; however, for batch normalization [11], we only use 86 samples to facilitate generalization. We precompute optical flow [32] before training and store the flow fields as JPEGs (with displacement vectors $> 20$ pixels clipped). During training, we use the same augmentations as in [1, 31]; i.e. randomly cropping from the borders and centre of the flow stack and sampling the width and height of each crop randomly within 256, 224, 192, 168, following by resizing to $224 \times 224$. The appearance stream is trained identically with a batch of 256 RGB frames and learning rate of $10^{-2}$ for $10K$ iterations, followed by $10^{-3}$ for another $10K$ iterations. Notably here we choose a very small batch size of 8 for normalization. We also apply random cropping and scale augmentations: We randomly jitter the width and height of the $224 \times 224$ input frame by $\pm 25\%$ and also randomly crop it from a maximum of $25\%$ distance from the image borders. The cropped patch is rescaled to $224 \times 224$ and passed as input to the network. The same rescaling and cropping technique is chosen to train the next two steps described below. In all our training steps we use random horizontal flipping and do not apply RGB colour jittering [14].

**ST-ResNet.** Second, to train our spatiotemporal ResNet we sample 5 inputs from a video with random temporal stride between 5 and 15 frames. This technique can be thought of as frame-rate jittering for the temporal convolutional layers and is important to reduce overfitting of the final model.

SGD is used with a batch size of 128 videos where 5 temporal chunks are extracted from each. Batch-normalization uses a smaller batch size of $128/32 = 4$. The learning rate is set to $10^{-3}$ and is reduced by a factor of 10 after $30K$ iterations. Notably, there is no pooling over time, which leads to temporal fully convolutional training with a single loss for each of the 5 inputs and both streams. We found that this strategy significantly reduces the training duration with the drawback that each loss does not capture all available information. We overcome this by the next training step.

**ST-ResNet\*.** For our final model, we equip the spatiotemporal ResNet with a temporal max-pooling layer after pool5 (see Table 1, temporal average pooling led to inferior results) and continue training as above with the learning rate starting from $10^{-4}$ for $2K$ iterations followed by $10^{-5}$. As indicated in Table 2, we now use 11 temporal chunks as input with the stride $\tau$ between these being randomly chosen from $[1, 15]$.

**Fully convolutional inference.** For fair comparison, we follow the evaluation procedure of the original two-stream work [20] by sampling 25 frames (and their horizontal flips). However, rather than using 10 spatial $224 \times 224$ crops from each of the frames, we apply fully convolutional testing both spatially (smallest side rescaled to 256) and temporally (the 25 frame-chunks) by classifying the video in a single forward pass, which takes $\approx$250ms on a Titan X GPU. For inference, we average the predictions of the fully connected layers (without softmax) over all spatiotemporal locations.

# 4  Evaluation

We evaluate our approach on two challenging action recognition datasets. First, we consider UCF101 [22], which consists of 13320 videos showing 101 action classes. It provides large diversity in terms of actions, variations in background, illumination, camera motion and viewpoint, as well as object appearance, scale and pose. Second, we consider HMDB51 [15], which has 6766 videos that show 51 different actions and generally is considered more challenging than UCF0101 due to the even wider variations in which actions occur. For both datasets, we use the provided evaluation protocol and report mean average accuracy over three splits into training and test sets.

## 4.1  Two-Stream ResNet with additive interactions

Table 3 shows the results of our two-stream architecture across the three training stages outlined in Sec. 3.4. For stream fusion, we always average the (non-softmaxed) prediction scores of the classification layer as this approach produces better results than averaging the softmax scores. Initially, let us consider the performance of the two streams, both initialized with ResNet50 models trained on ImageNet [8], but without cross-stream residual connections (2) and temporal convolutional layers (5). The accuracies for UCF101 and HMDB51 are 89.47% and 60.59%, (our HMDB51 motion stream is initialized from the UCF101 model). Comparatively, a VGG16 two-stream architecture produces 91.4% and 58.5% [1, 31]. In comparing these results it is notable that the VGG16 architecture is more computationally demanding (19.6 *vs*. 3.8 billion multiply-add FLOPs ) and also holds more model parameters (135M *vs*. 34M) than a ResNet50 model.

| Dataset | Appearance stream | Motion stream | Two-Streams | ST-ResNet | ST-ResNet* |
|---------|-------------------|---------------|-------------|-----------|------------|
| UCF101  | 82.29%            | 79.05%        | 89.47%      | 92.76%    | 93.46%     |
| HMDB51  | 43.42%            | 55.47%        | 60.59%      | 65.57%    | 66.41%     |

Table 3: Classification accuracy on UCF101 and HMDB51 in the three training stages of our model.

We now consider our proposed spatiotemporal ResNet (ST-ResNet), which is initialized by our two-stream ResNet50 model of above and subsequently equipped with 4 residual connections between the streams and 16 transformed temporal convolution layers (initialized as averaging filters). The model is trained end-to-end with the loss layers unchanged (we found that using a single, joint softmax classifier overfits severely to appearance information) and learning parameters chosen as in Table 2. The results are shown in the penultimate column of Table 3. Our architecture significantly improves over the two-stream baseline indicating the importance of residual connections between the streams as well as temporal convolutional connections over time. Interestingly, research in neuroscience also suggests that the human visual cortex is equipped with connections between the dorsal and the ventral stream to distribute motion information to separate visual areas [3, 27]. Finally, in the last column of Table 3 we show results for our ST-ResNet* architecture that is further equipped with a temporal max-pooling layer to consider larger temporal windows in training and testing. For training ST-ResNet* we use 11 temporal chunks at the input and the max-pooling layer pools over 5 chunks to expand the temporal receptive field at the loss layer to a maximum of 705 frames at the input. For

testing, where the network sees 25 temporal chunks, we observe that this long-term pooling further improves accuracy over our ST-ResNet by around 1% on both datasets.

## 4.2 Comparison with the state-of-the-art

We compare to the state-of-the-art in action recognition over all three splits of UCF101 and HMDB51 in Table 4 (left). We use ST-ResNet*, as above, and predict the videos in a single forward pass using fully convolutional testing. When comparing to the original two-stream method [20], we improve by 5.4% on UCF101 and by 7% on HMDB51. Apparently, even though the original two-stream approach has the advantage of multitask learning (HMDB51) and SVM fusion, the benefits of our deeper architecture with its cross-stream residual connections are greater. Another interesting comparison is against the two-stream network in [18], which attaches an LSTM to a two-stream Inception [23] architecture. Their accuracy of 88.6% is to date the best performing approach using LSTMs for action recognition. Here, our gain of 4.8% further underlines the importance of our architectural choices.

| Method | UCF101 | HMDB51 | Method | UCF101 | HMDB51 |
|---|---|---|---|---|---|
| Two-Stream ConvNet [20] | 88.0% | 59.4% | IDT [29] | 86.4% | 61.7% |
| Two-Stream+LSTM[18] | 88.6% | - | C3D + IDT [26] | 90.4% | - |
| Two-Stream (VGG16) [1, 31] | 91.4% | 58.5% | TDD + IDT [30] | 91.5% | 65.9% |
| Transformations[31] | 92.4% | 62.0% | Dynamic Image Networks + IDT [2] | 89.1% | 65.2% |
| Two-Stream Fusion[5] | 92.5% | 65.4% | Two-Stream Fusion[5] | 93.5% | 69.2% |
| ST-ResNet* | **93.4%** | **66.4%** | ST-ResNet* + IDT | **94.6%** | **70.3%** |

Table 4: Mean classification accuracy of the state-of-the-art on HMDB51 and UCF101 for the best ConvNet approaches (left) and methods that additionally use IDT features (right). Our ST-ResNet obtains best performance on both datasets.

The Transformations [31] method captures the transformation from start to finish of a video by using two VGG16 Siamese streams (that do not share model parameters, i.e. 4 VGG16 models) to discriminatively learn a transformation matrix. This method uses considerably more parameters than our approach, yet is readily outperformed by ours. When comparing with the previously best performing approach [5], we observe that our method provides a consistent performance gain of around 1% on both datasets.

The combination of ConvNet methods with trajectory-based hand-crafted IDT features [29] typically boosts performance nontrivially [2, 26]. Therefore, we further explore the benefits of adding trajectory features to our approach. We achieve this by simply averaging the L2-normalized SVM scores of the FV-encoded IDT descriptors (i.e. HOG, HOF, MBH) [29] with the L2-normalized video predictions of our ST-ResNet*, again without softmax normalization. The results are shown in Table 4 (right) where we observe a notable boost in accuracy of our approach on HMDB51, albeit less on UCF101. Note that unlike our approach, the other approaches in Table 4 (right) suffer considerably larger performance drops when used without IDT, e.g. C3D [26] reduces to 85.2% on UCF101, while Dynamic Image Networks [2] reduces to 76.9% on UCF101 and 42.8% on HMDB51. These relatively larger performance decrements again underline that our approach is better able to capture the available dynamic information, as there is less to be gained by augmenting it with IDT. Still, there is a benefit from the hand-crafted IDT features even with our approach, which could be attributed to its explicit compensation of camera motion. Overall, our 94.6% on UCF101 and 70.3% HMDB51 clearly sets a new state-of-the-art on these widely used action recognition datasets.

## 5 Conclusion

We have presented a novel spatiotemporal ResNet architecture for video-based action recognition. In particular, our approach is the first to combine two-stream with residual networks and to show the great advantage that results. Our ST-ResNet allows the hierarchical learning of spacetime features by connecting the appearance and motion channels of a two-stream architecture. Furthermore, we transfer both streams from the spatial to the spatiotemporal domain by transforming the dimensionality mapping filters of a pre-trained model into temporal convolutions, initialized as residual filters over time. The whole system is trained end-to-end and achieves state-of-the-art performance on two popular action recognition datasets.

**Acknowledgments.** This work was supported by the Austrian Science Fund (FWF) under project P27076 and NSERC. The GPUs used for this research were donated by NVIDIA. Christoph Feichtenhofer is a recipient of a DOC Fellowship of the Austrian Academy of Sciences at the Institute of Electrical Measurement and Measurement Signal Processing, Graz University of Technology.

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
