[Reviews · NeurIPS 2016]

Reviewer 1

Summary

In this paper, authors use ResNets architecture to learn human actions in a two-stream ConvNet architecture. They changes two-stream network in various ways: 1) Using ResNets architecture in the two-stream base-line method 2) Introducing residual connections between motion and appearance streams and also increasing the effective temporal receptive field by adding learnable residual connections over time. The authors also realized by practice that the performance of the method could be boosted using sub-batch normalization instead of batch normalization, frame rate jittering, and applying temporal max-pooling just after pool5 layer.

Qualitative Assessment

ResNets and two-stream networks are two main ingredients of this paper. While, authors successfully have identified and used the strong aspects of these methods and applied it to their application, the paper lacks strong novelty. Since there are multiple practical ideas to boost the performance of the method, it requires more experiments to clarify the effect of each part separately. For instance, it is not clear how much of the final performance is due to the sub-mini batch method and if the method could still beat the state-of- the-art without using it. Beside this, without any intuition, using sub-mini batch method is not reliable. I am worried that it only shows some improvements in the few datasets that are tried in the paper but not be a good idea in general. It needs intuition to show it effectiveness, and also more experiments should be done to see whether or not it is a dataset specific, application specific. or a general result. While adding skip connection between two streams seems more arbitrary, adding residual connections across time method seems more interesting to me. Unfortunately, the details provided in the later method is not explored enough and explained in details. For example, is the skip connections removed in the batch boundaries in forward and backward passes? If yes how it might effect the predictions near the boundaries of the batches? After rebuttal: I believe the authors' responses in terms of novelty are convincing hence updating my scores to 3. However, I still believe it would be good to show the specific ablation study for the different design components in the paper (sub-batch normalization, skip connection and temporal residual).

Confidence in this Review

2-Confident (read it all; understood it all reasonably well)


Reviewer 2

Summary

The paper extends the two-stream convolutional network architecture of [19] by introducing residual connections between the appearance and motion streams. In addition it also shows how pre-trained image ConvNets can be transformed into spatio-temporal networks by transforming the spatial dimensionality mapping filters in the residual paths to temporal filters. The approach is evaluated on UCF101 and HMDB datasets and comparisons show the proposed approach obtains state-of-art results on both datasets.

Qualitative Assessment

Positives - The paper shows how to effectively use residual connections in the spatio-termporal setting needed for action recognition. - Results on two important action recognition datasets validates the proposed approach. Negatives - The overall novelty is only incremental. - Section 3.2 is probably one of the key contributions of the paper and could be expanded and explained better.

Confidence in this Review

3-Expert (read the paper in detail, know the area, quite certain of my opinion)


Reviewer 3

Summary

The paper presents a novel architecture that 1) combines residual networks with two-stream convolutional networks, and 2) injects connections from the motion stream to the appearance stream, to be able to capture spatio-temporal features. The paper shows experiments in both of the main action recognition datasets, achieving state-of-the-art accuracy in both.

Qualitative Assessment

Overall I think the paper is great: good idea, careful experimentation, great results and clearly written. Although the basic components of the architecture are pre-existing, I think that the high performance and careful experimentation and description make it a very useful contribution. I only miss some experiments to visualize what kinds of spatio-temporal features are being learned. Since large temporal windows are important (278-280) I would add a relevant reference: Long-term Temporal Convolutions for Action Recognition, Gul Varol, Ivan Laptev, Cordelia Schmid Small typos: (L 187) "resp." (L 255) "UCF0101"

Confidence in this Review

2-Confident (read it all; understood it all reasonably well)


Reviewer 4

Summary

The paper proposes an extension of the two-stream approach to action recognition. It builds on top of a number of recent advances in designing Convolutional networks, most notably 1x1 dimensionality reduction and residual connections. Authors propose two modifications of the two-stream network - residual connection from motion to appearance stream and residual temporal filtering. They use these techniques to advance the state-of-the-art on two datasets, most notably on HMDB51.

Qualitative Assessment

The paper is well written and structured. It is mostly easy to read and follow the argument. Its novelty does not lie within inventing new techniques, but rather in creatively using the known ones. I'm not sure about the architecture's potential impact - one needs to access it on a bigger data, but it does look promising. A few points to improve: First, Table 3 shows results of the baseline ResNet50 and the proposed model with both modifications enabled. It would be good to see the contribution of motion-appearance connections and temporal filtering separately. Second, the section about temporal filtering is somewhat unclear. I've reread it multiple times and still am not sure whether I understand it correctly. It would be really helpful to expand it a bit. Third, it would be really nice to see the result of this architecture on the Sports-1m dataset. However, I do understand that this dataset is huge and very computationally expensive to handle. Future work could address this, maybe?

Confidence in this Review

2-Confident (read it all; understood it all reasonably well)


Reviewer 5

Summary

This paper proposed a deep CNN architecture for video action recognition by extending residual networks. The proposed architecture combines both spatial stream and temporal stream into a single network, and extends to model longer period of temporal information. This work achieves state-of-the-art performance on two public action recognition datasets.

Qualitative Assessment

This paper presents a framework that improves two stream networks for video action recognition by extending residual network to combine information from two streams into one single network. It significantly improves over previous state-of-the-art on two popular video action recognition benchmark. The downside of this paper is the limited novelty. There are previous work tried to combine two streams into a single network [1,2], and the temporal convolution is not new either [3]. Although the way to combine two streams is slightly different from previous work, the proposed approach is still pretty straightforward. Since this paper mainly consists of incremental ideas on previous work, more details could be given on how those ideas help in solving the tasks to gain more insights. For example, separate ablation studies would be useful to understand how the techniques in section 3.1 and 3.2 help. The temporal convolution described in section 3.2 is just replacing spatial convolution with temporal convolution, which is not really related to residual connections. The term “residual” could be dropped in the title of section 3.2 and line 166. Some details in the paper are unclear: - Are the HMDB51 networks initialized from UCF101 networks as in previous work? - The proposed network sees 705 frames (line 193). How do you handle videos with less than 705 frames? - Line 233: “Batch-normalization uses a smaller batch size of 128/32.” Does that mean the batch size is 128/32 = 4 as discussed above, or the batch size is 128 or 32? This is a bit confusing. - Is the loss of ST-ResNet defined as the sum of loss of both streams? Typo: Line 255: UCF0101 -> UCF101 Overall, this paper is clearly written, and produces very good results on video action recognition datasets. [1] C. Feichtenhofer, A. Pinz, A. Zisserman. Convolutional Two-Stream Network Fusion for Video Action Recognition. https://arxiv.org/abs/1604.06573 [2] E. Park, X. Han, T. L. Berg, A. C. Berg. Combining Multiple Sources of Knowledge in Deep CNNs for Action Recognition. WACV 2016 [3] L. Pigou, A. van den Oord, S. Dieleman, M. Van Herreweghe, J. Dambre, Beyond Temporal Pooling: Recurrence and Temporal Convolutions for Gesture Recognition in Video. http://arxiv.org/abs/1506.01911

Confidence in this Review

2-Confident (read it all; understood it all reasonably well)